# Dermatoglyphics as a Risk Indicator for Anterior Cruciate Ligament Injuries in Futsal Athletes

**DOI:** 10.3390/jfmk10040399

**Published:** 2025-10-15

**Authors:** Ben Hur Soares, Rudy José Nodari Júnior, Estélio Henrique Martin Dantas, Arnaldo Tenório da Cunha, Adriano Pasqualotti

**Affiliations:** 1Institut of Helht, University of Passo Fundo, Passo Fundo 99052-900, Brazil; 2Research Department, Salus Dermatoglifia, Luzerna 89609-000, Brazil; rudy.nodarijunior@salusdermatoglifia.com.br; 3Postgraduate Program in Nursing and Biosciences (PPGEnfBio), Federal University of Rio de Janeiro, Rio de Janeiro 21941-901, Brazil; estelio.dantas@unirio.br; 4Kinanthropometry, Physical Activity and Health Promotion Laboratory (LACAPS), Federal University of Alagoas, Arapiraca 57309-005, Brazil; arnaldo.junior@arapiraca.ufal.br; 5Graduate Program in Human Aging (PPGEH), University of Passo Fundo, Passo Fundo 99052-900, Brazil; pasqualotti@upf.br

**Keywords:** risk factors, soccer, knee injuries, athletic injuries, athletes

## Abstract

**Background:** Identifying factors that predispose futsal athletes to anterior cruciate ligament (ACL) injuries is crucial for developing effective prevention strategies. This study aimed to determine whether specific dermatoglyphic markers are associated with an increased risk of ACL injury in this population. **Methods:** This retrospective case–control study analyzed 212 former male futsal athletes, divided into an injury group (n = 85 with a history of ACL injury) and a control group (n = 127 without injury). Fingerprint patterns (arches, loops, and whorls) and quantitative line counts were collected and analyzed using the dermatoglyphics method. Chi-square tests and log-linear regression were used for statistical analysis. **Results:** While no significant differences were found in the quantitative line counts between groups (*p* > 0.05), a significant association was identified for specific fingerprint patterns. The spiral whorl (WS) pattern on the left index finger (*p* = 0.043) and the right little finger (*p* = 0.007) was significantly more frequent in the ACL injury group. Overall, athletes presenting the WS pattern had approximately twice the odds of having a history of ACL injury (OR = 2.028, 95% CI 1.493–2.756). **Conclusions:** The findings suggest that specific dermatoglyphic patterns, particularly the spiral whorl, may serve as an indicator of a potential biological predisposition to ACL injuries in futsal athletes. This finding suggests dermatoglyphics could be a potential component for future multifactorial risk assessment models in futsal.

## 1. Introduction

Futsal, an indoor variant of football [1,2], has grown into a popular global sport [3,4,5,6,7]. Characterized by its dynamic nature that requires diverse physical abilities [8,9], it is played by over 270 million people worldwide [10]. This has prompted sports science to develop methods for improving athletic performance and results [11]. Consequently, enhancing functional performance is crucial for futsal athletes, not only to improve their success but also to reduce the risk of injuries [12]. Injuries are observed to occur because of the movement patterns required in this sport, such as jumps, short and long displacements, rapid changes in direction, technical actions, and frequent physical contact between players [13,14,15]. Factors, such as age, training load, level of play, tactical dynamics, and training patterns may contribute to the occurrence of injuries [16]. These considerations are also crucial for women’s futsal, as female athletes may experience different joint stresses, prompting specific prevention protocols [17,18,19,20].

Musculoskeletal injuries are frequent in futsal, with anterior cruciate ligament (ACL) tears being among the most severe, accounting for approximately 79% of all joint injuries [17,18,19]. These ruptures are common in both amateur and professional athletes [20] and are typically associated with non-contact mechanisms (≈70%) such as jumping, sudden deceleration, and pivoting movements [21,22,23,24,25,26]. The consequences are significant, including a recovery period of six to nine months and a high rate of career abandonment due to physical trauma, which affects up to 47% of professional players [27,28,29]. These events negatively impact an athlete’s physical and mental health long after their career ends [20,30,31,32].

While biomechanical events represent modifiable risk factors, there is also significant potential for a non-modifiable genetic predisposition to these injuries [33]. This study explores this genetic link through dermatoglyphics, the analysis of fingerprints, which are immutable dermal traits formed concurrently with the musculoskeletal system during gestation [34,35]. The patterns are broadly classified into three basic shapes—arch (A), loop (L), and whorl (W) [36,37,38]—and have been successfully used for general health prognosis and for profiling innate athletic potential in high-performance sports [[39],[40],[41],[42],,[43],[44],[45],[46],[47],[48],[49]].

However, the specific association between these genetic markers and the susceptibility to musculoskeletal injuries like ACL tears remains largely unexplored. Confirming such a link could establish a novel, non-invasive tool for early risk stratification. Therefore, this study aims to determine whether specific markers of biological individuality in futsal athletes can be used to identify an increased risk of ACL injuries.

## 2. Materials and Methods

### 2.1. Design

This exploratory-analytical study was conducted following a retrospective design, approved by the Research Ethics Committee of the University of Passo Fundo, Brazil (Protocol No. 4.870.728). The research protocol involved two primary instruments: the dermatoglyphic method for analyzing biological markers and a structured questionnaire focusing on the athletes’ sports careers and injury history. To ensure objectivity and mitigate potential bias, a double-blinding procedure was implemented. The researchers responsible for collecting the dermatoglyphic data were blinded to the participants’ ACL injury status. Subsequently, the statistician responsible for data analysis was also blinded to the group allocations, receiving only coded data to perform the statistical modeling and comparisons. All collected information was handled confidentially to protect the participants’ identities.

### 2.2. Participants

Participants were recruited through a non-probabilistic sampling method from a network of former professional futsal athletes. The researchers made initial contact via messaging applications and email, outlining the study’s objectives and procedures. The final sample consisted of 212 former male futsal athletes from Brazil, Spain, and Italy, all athletes played in national and international leagues. These participants were allocated into two distinct groups: an injury group comprising 85 athletes with a confirmed history of ACL injury and a control group of 127 athletes with no history of ACL injury. The cohort was composed of 82.5% white, 11.0% brown, and 6.5% black individuals, with 77.9% being right-handed. The mean age at which the athletes concluded their professional careers was 39.1 ± 9.0 years, and the mean age for starting specific futsal training was 9.4 ± 3.8 years. All individuals voluntarily agreed to participate and provided written informed consent prior to data collection.

### 2.3. Protocol

The protocol for identifying markers of biological individuality was based on the dermatoglyphics method established by Cummins and Midlo [38]. To capture, process, and analyze the markers, we used the method validated for the Brazilian population [41,50]. Data collection was conducted using a Watson Mini digital biometric reader, which digitizes fingerprints through a rolling scan process. Each participant rolled their distal phalanges from the ulnar to the radial side on the scanner to ensure a complete capture of the print patterns. This digital method has been previously validated for accuracy against traditional ink-based techniques. The collected images were then subjected to both qualitative and quantitative analysis, as illustrated in Figure 1. For the qualitative analysis, each fingerprint was classified into one of five patterns: Arch (A), characterized by the absence of deltas; Ulnar Loop (LU) and Radial Loop (LR), both possessing a single delta; and Whorl (W) and Spiral Whorl (WS), which are identified by the presence of two deltas. For the quantitative analysis, two metrics were determined. First, the total delta count (D10) was calculated by summing the deltas from all ten fingers. Second, the line count for each finger was determined by tracing a straight line (Galton’s line) from the delta to the core (or nucleus) of the pattern and counting the number of ridges intersected. These individual counts were then summed to produce the Sum of Quantitative Total Lines for the Left Hand (SQTLE), the Right Hand (SQTLD), and the Overall Sum (SQTL). For coding purposes, hands were designated MESQL (Left) and MDSQL (Right), while fingers were numbered D1 (thumb) to D5 (little finger).

### 2.4. Statistical Analysis of Data

A sensitivity power analysis was conducted to confirm the adequacy of the sample size. This post hoc analysis confirmed that the final sample (85 cases, 127 controls) was sufficiently robust to detect the observed differences. Data distribution normality was assessed using the Kolmogorov–Smirnov test, and homoscedasticity of variances was checked with Levene’s test. A Student’s *t*-test was used to compare means, and the chi-square test determined associations among print patterns. Effect sizes were calculated (Cohen’s d for *t*-tests; Odds Ratio for regression) to determine the magnitude of the findings. For the ten individual finger analyses, a Bonferroni correction was applied to adjust for multiple comparisons, setting the significance threshold at *p* < 0.005.

Furthermore, a binary logistic regression was performed to assess the predictive power of the dermatoglyphic markers while controlling for the potential confounding effect of age at retirement. In this model, the ACL injury status (present/absent) served as the dependent variable, with the primary fingerprint patterns and age included as independent predictors. All statistical analyses were conducted using Jamovi (version 2.6) and R Language (version 4.4) (Appendix B), with a general significance level (α) of 0.05.

## 3. Results

Figure 2 shows the results of comparison of the age of athletes with and without ACL injuries when they stopped playing professionally.

The mean age at which athletes retired from their professional careers showed a statistically significant difference between the groups (t = −2.660, *p* = 0.008), with a small-to-medium effect size (Cohen’s d = −0.373). The group without a history of ACL injury retired, on average, at a later age (41.1 ± 7.7 years) compared to the group with a history of ACL injury (38.0 ± 9.5 years).

The log-linear model showed a significant association between the presence of ACL injuries and fingerprint patterns, as detailed in the omnibus likelihood ratio test (Table 1) and the model’s coefficients (Table 2). Figure 3 visually illustrates these differences, presenting the estimated marginal means of fingerprint patterns for the groups with and without ACL injuries. Table 3 shows the results of comparing the number of lines per finger and hand for the presence and absence of ACL injuries.

**Figure 3 jfmk-10-00399-f003:**
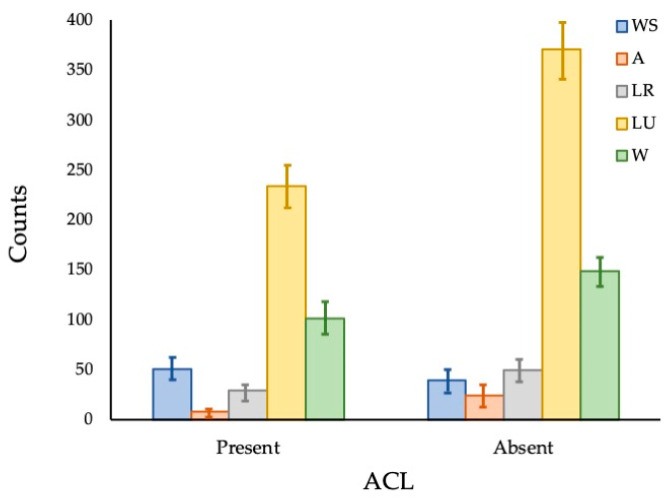
Estimated marginal means of presence and absence of ACL injuries versus print patterns. Modeling of the categorical variables: presence and absence of ACL injuries and print patterns; log-linear regression; statistically significant difference for *p* < 0.05.

The mean values of the number of lines did not show statistically significant differences (*p* > 0.05) for the groups with and without the injuries. Table 4 shows the results of the association between the print patterns per hand for the presence and absence of anterior cruciate ligament injuries.

There was a statistically significant association between the print pattern for both the left hand (χ^2^ = 915.072; df = 4; *p* = 0.005) and the right hand (χ^2^ = 18.015; df = 4; *p* = 0.001). There was also an association between the global analysis of hands and print pattern (χ^2^ = 27.125; df = 4; *p* < 0.001). The radial print pattern of the left hand, the ulnar loop print pattern of the right hand, and the radial loop print pattern of the global analysis of hands were associated with the absence of injuries; the spiral whorl (WS) print pattern was associated with the presence of ACL injuries in the three comparisons performed (right hand, left hand, and global analysis of hands). Table 5 presents the association between fingerprint patterns per finger and ACL injuries.

There was a statistically significant association between the spiral whorl (WS) print pattern of the index finger of the left hand (MED2|χ^2^ = 9.875; df: 4; *p* = 0.043) in the group with ACL injuries. There was also an association for the whorl (W) print pattern and spiral whorl (WS) print pattern for the little finger of the right hand (MED5) (χ^2^ = 11.978; df: 3; *p* = 0.007). The group with ACL injuries had a greater number of spiral whorl (WS) print patterns; the group without injuries had a greater number of whorl (W) print patterns. Table 6 shows the spiral whorl (WS) print pattern and the definite diagnosis for the presence and absence of anterior cruciate ligament injuries.

The sensitivity was 0.120 (see Appendix A for all diagnostic test formulas), indicating whether the presence of the WS print pattern can indicate a higher risk of ACL injuries (disease/condition) in futsal athletes. The specificity was 0.937 and indicates whether the presence of the WS print pattern can be used to rule out ACL injuries (disease/condition) in futsal athletes. Figure 4 shows the cutoff values between the coefficients for specificity and sensitivity.

The accuracy (0.609) determined the proportion of all correct tests for the presence of WS print pattern versus the presence of ACL injuries and the absence of WS print pattern versus the absence of ACL injuries (true positives and true negatives) across all results obtained. A positive predictive value (PPV) of 0.560 indicated the probability that a futsal athlete with a positive test has ACL injuries. A negative predictive value (NPV) of 0.614 indicated the probability that a futsal athlete with a negative test does not have the condition. The likelihood ratio for a positive test was 1.905 and indicated the likelihood of a positive test in an athlete with ACL injuries compared to an athlete without ACL injuries. The likelihood of having a confirmed ACL injury history when the WS print pattern (102/80) is detected was 1.275; the likelihood of not having ACL when the WS print pattern (748/1190) is detected was 0.629. The odds ratio (OR) for the presence of a confirmed ACL injury history versus its absence (1.275/0.629) was 2.028 (95% CI = (1.493; 2.756). Given that the 95% CI of OR does not include a value of 1, futsal athletes with clinical examination findings suggestive of a confirmed ACL injury and that show a WS print pattern are approximately 1.5 to 2.8 times more likely to have ACL injuries than those that do not show a WS print pattern. To assess the predictive power of dermatoglyphic markers while controlling for the potential confounding effect of retirement age, a binary logistic regression analysis was performed. The model was constructed to predict the dependent variable (ACL injury status: present or absent) from the independent variables: age at retirement and the frequency of the Spiral Whorl (WS) and Arch (A) fingerprint patterns. Table 7 below presents the model fit statistics, while Table 8 details the coefficients and Odds Ratios for each predictor.

To address the potential confounding effect of the age of retirement on the findings, a logistic regression model was performed, including both the spiral whorl (WS) pattern and age as predictors of ACL injury status. In this model, the association between the presence of the WS pattern and a history of ACL injury remained statistically significant (*p* = 0.013), even after controlling for the influence of age. This indicates that the dermatoglyphic marker is an independent predictor, distinct from the age at which an athlete concludes their career.

## 4. Discussion

We have divided this section into three subsections: (a) Dermatoglyphics as a risk indicator, career time, and retirement age of the athlete; (b) number of lines defined by dermatoglyphics; (c) print standards. In all these subsections, we present a concise and precise description of the experimental results found, the interpretations we made on these results, and the experimental conclusions we drew from the results found.

### 4.1. Dermatoglyphics as a Risk Indicator, Career Time, and Retirement Age of the Athlete

Risk assessment based on dermatoglyphics may provide a robust tool for prior observation of genetically predisposed diseases. Genetic studies can provide an additional method of prediction and help prevent potential health problems [42,44,51,52]. Each organism is unique and has an epigenetic trait inherited and generated during fetal development in the womb [53]. The definition of print patterns is closely related to the functioning of the central nervous system. The markers defined at this stage of development can be used as a simple and practical method for the prognosis of health conditions [54]. The markers of biological individuality can enable the discovery of the innate potential of the individual [41]. The combinations of genetic variants and markers of biological individuality identified by dermatoglyphic analysis can be used to evaluate the risk of ACL injuries. Our study aimed at identifying dermatoglyphic markers (lines and print patterns) that can be associated with a history of ACL injuries.

Athletes in most sports have a relatively short duration of career [55]. Career transition refers to the point in time when the athlete prepares to stop training and competing. The end of a sports career has an impact on the personal lifestyle of a former athlete [56]. The former athlete must adapt to new life conditions, assuming different roles that are not necessarily related to the activity performed in the past [57]. Depending on the sport that the athlete practices, their athletic career can last between 15 and 25 years [58]. The end of the athlete’s career occurs at 35.7 ± 3.83 years on mean [55]. The mean age determined in our study was higher and was 41.1 ± 7.7 years for the group with ACL and 38.0 ± 9.5 years for the group without ACL.

### 4.2. Number of Lines Defined by Dermatoglyphics

We did not find statistically significant differences between the number of lines in the groups with and without ACL injuries. This finding is comparable to the number of lines found by fingerprints in women with breast cancer [43]. The mean total number of lines (TNL) showed no statistically significant difference (t = 0.515; *p* = 0.581) between the group with ACL injuries (121.3 ± 35.9 total lines) and the group without ACL injuries (118.6 ± 38.8 total lines). These results are like the dermatoglyphic markers found in male high-level futsal athletes (124.6 ± 40.8 total lines) [52]. The results of our study are like the study that analyzed the number of lines in female high-level futsal athletes (121.7 ± 39.2 total lines) [51]. In these two studies, the mean number of lines was significantly different. However, the authors compared the mean number of lines in high-performance athletes with individuals who play the same sport but are not high-performance athletes.

Another study found a statistically different meaning number of lines in the MESQL5 and MDSQL4 fingers [59,60]. The mean number of lines was higher in the group “high physical fitness level” than in the group “low physical fitness level”. In a study that investigated the markers of biological individuality as a mechanism for the prognosis of heart diseases, the authors found that the mean number of lines on the MDSQL5 finger was significantly higher in the group with heart disease than in the control group [61]. In our study, the results showed no statistically significant differences (t = 0.943; *p* = 0.347) between the mean values of the number of lines in the group with ACL injuries (12.2 ± 4.8 lines) and in the group without ACL injuries (11.9 ± 4.7 lines). The results of our study are like those of a study analyzing the motor ability and speed in children and adolescents [62].

### 4.3. Print Patterns

We found statistically significant differences between the print patterns in the index finger of the left hand (MED2|χ^2^ = 9.875; df = 4; *p* = 0.043) and the little finger of the right hand (MED5|χ^2^ = 11.978; df: 3; *p* = 0.007). In both fingers (MED2 and MED5), the group with injuries showed a statistically significant association in the number of spiral whorl (WS) print patterns; in the group without ACL injuries, the whorl (W) print pattern was associated with the finger MED5. Considering the innate characteristics, the identification of fingerprint patterns that differ between the studied groups could be a determining factor for the prognosis of ACL injuries. In our study, regular practice, the number of games played, number of trips undertaken, and disciplined life of the athletes were observed in both groups. This shows that the phenotype behaves similarly in both groups. The identification of some specific markers of the genotype that are significantly different increases the likelihood that a futsal athlete will suffer an ACL injury.

In one study, a statistically significant association of the ulnar loop (LU) print pattern was found in all fingers of both hands in women diagnosed with breast cancer compared to the control group [44]. In our study, this pattern was more prevalent in the group of athletes without ACL injuries. The difference in the results, with one of the studies associating the marker with the group with the condition and our study indicating the absence of the condition, may suggest that there is a condition that we did not analyze and that potentiated the results found. However, it should be noted that the sample of the other study included only women, which could explain the difference in the association found.

The study that investigated the motor ability and speed in children and adolescents found a statistically significant association of the radial loop (LR) print pattern in the fingers of the left hand MED1 and MED5, and in the fingers of the right hand MDD1, MDD3, and MDD5 [62]. In our study, this print pattern showed no correlation between the analyzed groups.

## 5. Conclusions

This study concludes that specific dermatoglyphic markers are significantly associated with the incidence of anterior cruciate ligament injuries among former futsal athletes. The results demonstrate that the increased frequency of the spiral whorl (WS) pattern on the left index finger and the right little finger can serve as a biological indicator of a predisposition to ACL tears. These findings suggest that dermatoglyphics could be a valuable, non-invasive, and low-cost tool for the early identification of athletes at higher risk.

In practical terms, these findings could be implemented as a low-cost, non-invasive screening tool during pre-season assessments in elite sports environments. Athletes identified with at-risk dermatoglyphic markers, such as the spiral whorl (WS) pattern, would not be excluded but rather directed toward individualized and enhanced injury prevention programs. Such programs would emphasize neuromuscular control training, the refinement of landing and change-of-direction biomechanics, and targeted strengthening. Thus, dermatoglyphics would serve to stratify risk and personalize athlete management, complementing existing functional evaluation protocols.

However, the limitations of this study must be acknowledged. Our research focused exclusively on the association between dermatoglyphic markers and injury history, without concurrently evaluating well-established extrinsic and intrinsic risk factors. Variables such as training loads, fatigue, biomechanical factors (e.g., Q-angle, dynamic valgus knee, joint stability), and neuromuscular control were not included in our analysis. This omission prevents a multifactorial understanding of how genetic predispositions, indicated by dermatoglyphics, may interact with these known risk factors.

Therefore, future research should aim to integrate dermatoglyphic analysis with biomechanical assessments and training load monitoring to build more comprehensive and accurate predictive models. Expanding these investigations to include different ethnic groups, diverse performance levels, and intersex comparisons is also essential to determine the broader validity and applicability of these findings.

## Figures and Tables

**Figure 1 jfmk-10-00399-f001:**
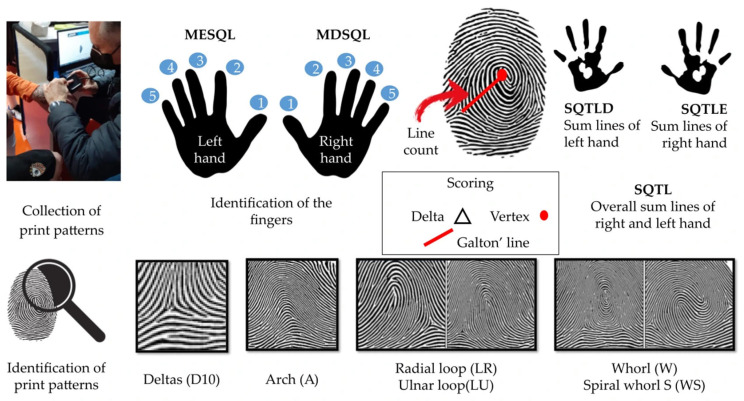
Demonstration of fingerprint collection, finger coding, marking of the points, and counting of the number of lines on the fingers.

**Figure 2 jfmk-10-00399-f002:**
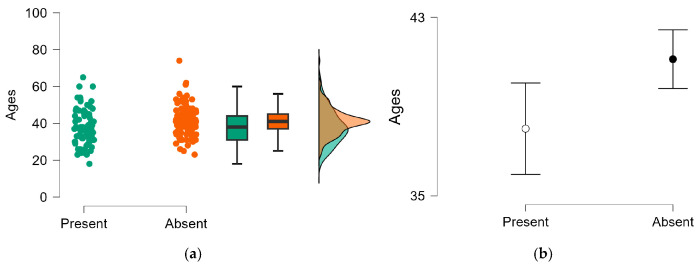
Athletes with and without ACL. Student’s *t*-test; statistically significant difference for *p* < 0.05. (**a**) Density of age distribution. (**b**) 95% CI mean of ages.

**Figure 4 jfmk-10-00399-f004:**
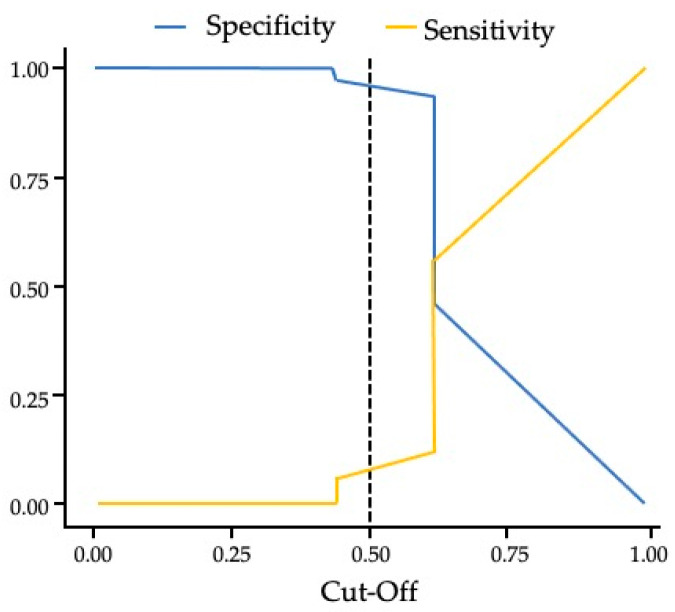
Cut-Off Plot. The dashed line indicates the 0.50 cut-off threshold.

**Table 1 jfmk-10-00399-t001:** Omnibus likelihood ratio test.

Predictor	χ^2^	df	*p*
ACL	1.605	1	0.205
Hands	0.050	1	0.823
Print patterns	606.254	4	<0.001
ACL ✻ Hands	0.028	1	0.867
Print patterns ✻ ACL	18.188	4	0.001
Hands ✻ Print patterns	1.836	4	0.766
ACL ✻ Hands ✻ Print patterns	6.221	4	0.183

Modeling of the categorical variables: presence and absence of ACL injuries, print patterns and hands; log-linear regression; statistically significant difference for *p* < 0.05. The ✻ symbol indicates an interaction term between the variables.

**Table 2 jfmk-10-00399-t002:** Coefficients of the log-linear regression model.

Predictor	Estimates	SE	Z	*p*
Intercept	3.664	0.160	22.879	<0.001
ACL:				
Present–Absent	0.268	0.213	1.261	0.207
Hands:				
Right hand–Left hand	0.050	0.224	0.224	0.823
Print patterns:				
A–WS	−0.445	0.256	−1.736	0.083
LR–WS	0.187	0.217	0.861	0.389
LU–WS	2.282	0.168	13.574	<0.001
W–WS	1.292	0.181	7.148	<0.001
ACL ✻ Hands:				
(Present–Absent) ✻ (Right hand–Left hand)	−0.050	0.299	−0.167	0.867
Print patterns ✻ ACL:				
(A–WS) ✻ (Present–Absent)	−1.408	0.459	−3.070	0.002
(LR–WS) ✻ (Present–Absent)	−1.228	0.349	−3.515	<0.001
(LU–WS) ✻ (Present–Absent)	−0.665	0.228	−2.921	0.003
(W–WS) ✻ (Present–Absent)	−0.713	0.252	−2.835	0.005
Hands ✻ Print patterns:				
(Right hand–Left hand) ✻ (A–WS)	−0.091	0.363	−0.250	0.802
(Right hand–Left hand) ✻ (LR–WS)	0.070	0.300	0.234	0.815
(Right hand–Left hand) ✻ (LU–WS)	−0.109	0.235	−0.464	0.642
(Right hand–Left hand) ✻ (W–WS)	0.050	0.252	0.200	0.841
ACL ✻ Hands ✻ Print patterns:				
(Present–Absent) ✻ (Right hand–Left hand) ✻ (A–WS)	0.091	0.649	0.140	0.889
(Present–Absent) ✻ (Right hand–Left hand) ✻ (LR–WS)	0.777	0.457	1.701	0.089
(Present–Absent) ✻ (Right hand–Left hand) ✻ (LU–WS)	−0.083	0.321	−0.259	0.796
(Present–Absent) ✻ (Right hand–Left hand) ✻ (W–WS)	0.157	0.350	0.449	0.653

Modeling of the categorical variables: ACL, print patterns and hands; arch: A, radial loop: LR, ulnar loop: LU, whorl: W, spiral whorl: WS; Log-linear regression; statistically significant difference for *p* < 0.05. The ✻ symbol indicates an interaction term between the variables.

**Table 3 jfmk-10-00399-t003:** Mean number of lines per finger and hand for the presence and absence of anterior cruciate ligament injuries.

Number of Lines	ACL	Mean	Standard Deviation	Standard Error	*t*/*p*
MESQL1	Present	13.9	5.5	0.6	−0.914
	Absent	14.6	5.1	0.46	0.362
MESQL2	Present	9.1	5.1	0.55	0.094
	Absent	9.1	5.5	0.49	0.925
MESQL3 *	Present	10.9	4.5	0.49	0.519
	Absent	10.5	5.5	0.49	0.604
MESQL4	Present	13.3	4.6	0.5	0.516
	Absent	12.9	5.5	0.49	0.607
MESQL5	Present	12.3	4.5	0.49	1.650
	Absent	11.3	4.7	0.42	0.101
MDSQL1	Present	15.9	5.2	0.56	0.442
	Absent	16.4	4.5	0.40	0.659
MDSQL2	Present	9.9	5.4	0.59	−0.796
	Absent	8.9	5.9	0.52	0.427
MDSQL3	Present	10.7	4.6	0.49	1.161
	Absent	10.5	5.0	0.45	0.247
MDSQL4 *	Present	13.2	4.6	0.50	0.199
	Absent	12.5	5.5	0.49	0.843
MDSQL5	Present	12.2	4.8	0.52	0.943
	Absent	11.9	4.7	0.42	0.347
SQTLE	Present	59.5	18.5	2.00	0.455
	Absent	58.3	20.1	1.79	0.650
SQTLD	Present	61.8	18.9	2.05	0.553
	Absent	60.3	19.9	1.77	0.581
SQTL	Present	121.3	35.9	3.90	0.515
	Absent	118.6	38.8	3.45	0.607
D10	Present	13.4	3.2	0.35	1.694
	Absent	12.6	3.5	0.31	0.092

Student’s *t*-test; statistically significant difference for *p* < 0.05. * indicates a violation of the assumption of homogeneity of variances Levene’s test. Levene’s test is significant (*p* < 0.05) indicating a violation of the assumption of homogeneity of variances.

**Table 4 jfmk-10-00399-t004:** Association between the print patterns per hand for the presence and absence of anterior cruciate ligament injuries.

Hands	ACL	Print Patterns					χ^2^|*p*
		A	LR	LU	W	WS	
Left hand	Present	8 (24.2%)	18 (27.7%)	257 (40.2%)	91 (39.1%)	51 (56.7%)	15.072
	Absent	25 (75.8%)	47 (72.3%)	382 (59.8%)	142 (60.9%)	39 (43.3%)	0.005
Right hand	Present	8 (25.0%)	42 (44.2%)	212 (37.1%)	112 (41.6%)	51 (55.4%)	18.015
	Absent	24 (75.0%)	53 (55.8%)	360 (62.9%)	157 (58.4%)	41 (44.6%)	0.001
Overall	Present	16 (24.6%)	60 (37.5%)	469 (38.8%)	203 (40.4%)	102 (56.0%)	27.125
	Absent	49 (75.4%)	100 (62.5%)	742 (61.2%)	299 (59.6%)	80 (44.0%)	<0.001

arch: A, radial loop: LR, ulnar loop: LU, whorl: W, spiral whorl: WS; Chi-square test; statistically significant difference for *p* < 0.05.

**Table 5 jfmk-10-00399-t005:** Association between the print patterns per finger for the presence and absence of anterior cruciate ligament injuries.

Finger Coding	ACL	Print Patterns					*p*
		A	LR	LU	W	WS	
MED1	Present	2 (33.3%)	3 (42.9%)	40 (38.1%)	18 (35.3%)	22 (51.2%)	0.559
	Absent	4 (66.7%)	4 (57.1%)	65 (61.9%)	33 (64.7%)	21 (48.8%)	
MED2	Present	4 (30.8%)	11 (26.8%)	33 (41.3%)	24 (40.7%)	13 (68.4%)	0.043
	Absent	4 (66.7%)	4 (57.1%)	65 (61.9%)	33 (64.7%)	21 (48.8%)	
MED3	Present	2 (22.2%)	1 (9.1%)	63 41.7%)	14 (43.8%)	5 (55.6%)	0.145
	Absent	7 (77.8%)	10 (90.9%)	88 (58.3%)	18 (56.3%)	4 (44.4%)	
MED2	Present	0 (0.0%)	1 (50.0%)	49 (38.9%)	28 (40.6%)	7 (63.6%)	0.250
	Absent	4 (100.0%)	1 (50.0%)	77 (61.1%)	41 (59.4%)	4 (36.4%)	
MED5	Present	0 (0.0%)	2 (50.0%)	72 (40.7%)	7 (31.6%)	4 (50.0%)	0.770
	Absent	1 (100.0%)	2 (50.0%)	105 (59.3%)	15 (68.2%)	4 (50.0%)	
MDD1	Present	2 (66.7%)	2 (66.7%)	31 (32.3%)	30 (43.5%)	20 (48.8%)	0.213
	Absent	1 (33.3%)	1 (33.3%)	65 (66.7%)	39 (56.5%)	21 (51.2%)	
MDD2	Present	2 (13.3%)	27 (45.0%)	23 (32.9%)	23 (46.9%)	10 (55.6%)	0.053
	Absent	13 (86.7%)	33 (55.0%)	47 (67.1%)	26 (53.1%)	8 (44.4%)	
MDD3	Present	3 (30.0%)	4 (33.3%)	60 (39.2%)	12 (41.4%)	6 (75.0%)	0.311
	Absent	7 (70.0%)	8 (66.7%)	93 (60.8%)	17 (58.6%)	2 (25.0%)	
MDD4	Present	1 (25.0%)	4 (40.0%)	33 (31.1%)	40 (44.9%)	7 (46.9%)	0.646
	Absent	3 (71.0%)	6 (60.0%)	61 (64.9%)	49 (55.1%)	8 (53.3%)	
MDD5 *	Present	0 (0.0%)	5 (50.0%)	65 (40.9%)	7 (21.2%)	8 (80.0%)	0.007
	Absent	0 (0.0%)	5 (50.0%)	94 (59.1%)	26 (78.8%)	2 (20.0%)	

arch: A, radial loop: LR, ulnar loop: LU, whorl: W, spiral whorl: WS; Chi-square test; statistically significant difference for *p* < 0.05. * There are no records of arch patterns on this finger (df = 3).

**Table 6 jfmk-10-00399-t006:** Association between the spiral whorl (WS) print pattern and the final diagnosis for the presence and absence of anterior cruciate ligament injuries.

WS Print Pattern	Definitive Diagnosis of ACL		Total
	Present	Absent	
Present	102	80	182
Absent *	748	1.190	1.938
Total	850	1.270	2.120

* Other patterns (A, LR, LU, W).

**Table 7 jfmk-10-00399-t007:** Logistic Regression Model Fit Summary.

Model Summary–Groups										
Model	Deviance	AIC	BIC	df	ΔΧ^2^	*p*	McFadden R^2^	Nagelkerke R^2^	Tjur R^2^	Cox & Snell R^2^
M_0_	285.5	287.518	290.875	211			0.000		0.000	
M_1_	270.1	278.086	291.512	208	15.432	0.001	0.054	0.095	0.072	0.070

Note. M_1_ includes Ages, WS, A.

**Table 8 jfmk-10-00399-t008:** Coefficients of the Logistic Regression Model for Predictors of ACL Injury.

						Wald Test			95% Confidence Interval	
									(Odds Ratio Scale)	
Model		Estimate	Standard Error	Odds Ratio	z	Wald Statistic	df	*p*	Lower Bound	Upper Bound
M_0_	(Intercept)	0.402	0.140	1.494	2.865	8.210	1	0.004	1.135	1.966
M_1_	(Intercept)	−1.043	0.721	0.352	−1.447	2.093	1	0.148	0.086	1.448
	Ages	0.042	0.018	1.042	2.348	5.512	1	0.019	1.007	1.079
	WS	−0.276	0.111	0.758	−2.490	6.199	1	0.013	0.610	0.943
	A	0.143	0.181	1.153	0.788	0.620	1	0.431	0.809	1.644

Note: Groups level ‘Absent’ coded as class 1.

## Data Availability

The dataset analyzed or generated during the study can be accessed: Paqualotti, Adriano; Soares, Ben Hur; Nodari Junior, Rudy José, 2023, “Data for Markers of Biological Individuality”, https://doi.org/10.7910/DVN/6ZFOZ7, Harvard Dataverse, V1, accessed on 19 September 2025.

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
