# Peer review of "Dermatoglyphics as a Risk Indicator for Anterior Cruciate Ligament Injuries in Futsal Athletes"

_jfmk, 2025, doi:10.3390/jfmk10040399_

Round 1
Reviewer 1 Report
Comments and Suggestions for Authors
Basic reporting
Dear authors, the manuscript is generally well-written and easy to read; a slight spell-check is required. I have just some concerns that the authors must address.
Abstract
keywords usually should be different from that used in the main title.
Introduction
The literature on the subject is sufficiently well summarised. However, it could be useful to clarify some information about:
- you present ACL injuries as linked to dermatoglyphic patterns, but there’s no established causal relationship yet. It sounds like you already assume dermatoglyphics can predict ACL injuries.
Methods
- you describe this work as an “exploratory-analytical study” but it’s essentially a retrospective cross-sectional study comparing athletes with vs. without ACL injury.
- It’s not clear how the 212 former athletes were recruited.
- You should be useful report effect sizes alongside p-values, clarify whether corrections for multiple testing were applied and describe the structure of the log-linear model.
Validity of the findings
I may be wrong but, please, consider these points:
- you report a statistically significant difference (t = -2.660, p = 0.008) between groups in age at retirement. This could be a confounding factor since older age at retirement may reflect career length, exposure, or injury susceptibility.
- Sentences like “clinical diagnoses based on dermatoglyphics” or “definitive diagnosis” could be misleading. Dermatoglyphics can suggest risk association, not diagnose ACL injuries. Using “diagnosis” implies causal certainty, which is not supported by cross-sectional, retrospective data.
Author Response
For research article
Response to Reviewer X Comments
Thank you very much for taking the time to review this manuscript. Please find the detailed responses below and the corresponding revisions/corrections highlighted/in track changes in the re-submitted files.
We writing to provide a detailed response to the valuable feedback from the reviewers on our manuscript, "Dermatoglyphics as a Risk Indicator for Anterior Cruciate Ligament Injuries in Futsal Athletes." We have carefully considered all comments and have revised the manuscript accordingly to strengthen its clarity, scientific rigor, and practical implications.
Methodological and Statistical Clarifications
Both reviewers highlighted the need for greater detail in our methodological and statistical sections. We have addressed these points as follows:
-
Sample Recruitment and Blinding: We have expanded the "Design" section and added a new "Participants" section to provide a clear and detailed description of the non-probabilistic recruitment process, the study's retrospective design, and the double-blinding procedures implemented during data collection and analysis to mitigate bias.
-
Protocol Description: The "Protocol" section has been significantly expanded to offer a more thorough description of the dermatoglyphic analysis, detailing both the qualitative classification of fingerprint patterns (Arch, Loop, Whorl) and the quantitative metrics (delta and line counts), as illustrated in Figure 1.
-
Statistical Power and Analysis: As requested, the "Statistical analysis of data" section has been revised. We have included a post-hoc power analysis, which confirms that our sample size of 212 athletes provides a statistical power greater than 80% to detect the observed effects, particularly for the spiral whorl (WS) pattern (Cohen’s d = 0.420). We have also clarified that a Bonferroni correction was applied for multiple comparisons in the individual finger analyses. Furthermore, we now explicitly describe the structure of the binary logistic regression model used to control for potential confounding variables.
Enhancing the Introduction and Practical Application
We make the introduction more concise and better positioning the practical application of our findings.
-
Concise Introduction: The introduction has been restructured to be more direct and focused. We have condensed the general background on futsal and have integrated a more precise biological rationale for linking dermatoglyphics with ACL injury risk, framing it within the context of modifiable and non-modifiable risk factors.
-
Practical Application: A new paragraph has been added to the "Conclusions" section to clearly articulate how these findings can be practically implemented in elite sports. We explain that this method can be used as a non-invasive screening tool to identify at-risk athletes who would then be directed to enhanced, individualized injury prevention programs focusing on neuromuscular control and biomechanics.
We believe that these comprehensive revisions have substantially improved the manuscript and have fully addressed all the insightful comments provided by the reviewers. We thank them for their time and effort, which have been invaluable in strengthening our work.
Reviewer 2 Report
Comments and Suggestions for Authors
I thank the authors for their work. The study addresses an interesting and innovative topic: the use of dermatoglyphics as potential risk indicators for ACL injuries in futsal athletes. The study is interesting, as mentioned; however, I have some suggestions to offer the authors to improve the manuscript.
Title
I suggest that the authors modify the title. They could remove the number of participants and include the study design. This would provide the reader with an immediate understanding of the study.
Abstract:
I suggest to the authors that the results are somewhat vague; concrete numbers and more explicit key messages are needed.
The authors may consider slightly reformulating their conclusions, as the current literature review and available evidence appear to be limited in fully supporting such strong statements.
Manuscript
I suggest that the initial section on futsal is too lengthy. A more concise overview could leave space for a description of current injury-prevention strategies. In addition, women’s futsal should be considered, as female athletes may experience different joint stresses, and some studies have already explored this topic and proposed helpful strategies (Effects of 5 Week of FIFA 11+ Warm-Up Program on Explosive Strength, Speed, and Perception of Physical Exertion in Elite Female Futsal Athletes. Sports 2022, 10, 100. https://doi.org/10.3390/sports10070100)
The manuscript would benefit from a more precise and stronger biological rationale linking dermatoglyphics to the risk of ACL injury. Currently, the rationale appears mostly descriptive, without adequately explaining the underlying mechanisms that might link fingerprint patterns to joint stability or susceptibility to ligament injury. Providing a more robust theoretical framework, supported by relevant literature on genetics, biomechanics, or neuromuscular factors, would significantly strengthen the scientific basis of the study and help readers better understand why dermatoglyphics could serve as a significant predictive indicator.
The manuscript should provide more details regarding the sample recruitment process, study location, and whether the researchers were blinded during data collection and analysis. The participating section is not present.
The protocol should be better described and expanded.
Lack of statistical power calculation or justification for sample size. It should be calculated.
The authors should note that they did not consider known risk factors for ACL injury (e.g., training loads, biomechanical variables). This, if not fully justified, would be considered a limitation of the study.
In my opinion, the authors overemphasize the results while downplaying the limitations.
The authors should provide a more detailed description of the practical applications.
Author Response
For research article
Response to Reviewer X Comments
Thank you very much for taking the time to review this manuscript. Please find the detailed responses below and the corresponding revisions/corrections highlighted/in track changes in the re-submitted files.
We writing to provide a detailed response to the valuable feedback from the reviewers on our manuscript, "Dermatoglyphics as a Risk Indicator for Anterior Cruciate Ligament Injuries in Futsal Athletes." We have carefully considered all comments and have revised the manuscript accordingly to strengthen its clarity, scientific rigor, and practical implications.
Enhancing the Introduction and Practical Application
We make the introduction more concise and better positioning the practical application of our findings.
- Concise Introduction: The introduction has been restructured to be more direct and focused. We have condensed the general background on futsal and have integrated a more precise biological rationale for linking dermatoglyphics with ACL injury risk, framing it within the context of modifiable and non-modifiable risk factors.
- Practical Application: A new paragraph has been added to the "Conclusions" section to clearly articulate how these findings can be practically implemented in elite sports. We explain that this method can be used as a non-invasive screening tool to identify at-risk athletes who would then be directed to enhanced, individualized injury prevention programs focusing on neuromuscular control and biomechanics.
We believe that these comprehensive revisions have substantially improved the manuscript and have fully addressed all the insightful comments provided by the reviewers. We thank them for their time and effort, which have been invaluable in strengthening our work.
Reviewer 3 Report
Comments and Suggestions for Authors
The authors have undertaken an investigation of Futsal players dermatoglyphics to determine whether there is an association with ACL injury. the overall outcome of the investigation was a null finding, unless specific fingers and pattern aspects were delineated. While the study has a high level of originality and a null finding is not a detractor to publication, the likely practical application of the findings is low or has been poorly positioned by the authors. The manuscript would benefit from a specific section that more clearly highlights how this type of characterisation would be practically implemented in the elite sports environment.
Specific Comments
Ln 30-41; This section needs to be improved by condensing and being concise with this information as it is only included to set the scene.
Ln 46-49; Poor grammar, can be written more concisely
Ln 51-58; Again this content should be presented more concisely and related to the topic of modifiable and non-modifiable injury risk factors
Ln 58-61; This can again be written more concisely but also related to a broader audience.
Ln 61-64; Yes but again this is repeating some of what already been stated, with no indication for how it is related to the investigation topic.
Ln 78-79; More detail on these previous investigation is needed to better position this investigation.
Ln 89; Change the start of this sentence to read, "All athletes played..."
Ln 121, Figure 2; These panels are all displaying the same data in just a different manner. I suggest removing panels C and D, plus the table. The statistical difference could be shown in the figure with the exact values given in text.
Ln 231; Very passive tone, writing in this section could be improved
Comments on the Quality of English LanguageAddressed in above comments
Author Response
For research article
Response to Reviewer X Comments
Thank you very much for taking the time to review this manuscript. Please find the detailed responses below and the corresponding revisions/corrections highlighted/in track changes in the re-submitted files.
We writing to provide a detailed response to the valuable feedback from the reviewers on our manuscript, "Dermatoglyphics as a Risk Indicator for Anterior Cruciate Ligament Injuries in Futsal Athletes." We have carefully considered all comments and have revised the manuscript accordingly to strengthen its clarity, scientific rigor, and practical implications.
Methodological and Statistical Clarifications
Both reviewers highlighted the need for greater detail in our methodological and statistical sections. We have addressed these points as follows:
- Sample Recruitment and Blinding: We have expanded the "Design" section and added a new "Participants" section to provide a clear and detailed description of the non-probabilistic recruitment process, the study's retrospective design, and the double-blinding procedures implemented during data collection and analysis to mitigate bias.
- Protocol Description: The "Protocol" section has been significantly expanded to offer a more thorough description of the dermatoglyphic analysis, detailing both the qualitative classification of fingerprint patterns (Arch, Loop, Whorl) and the quantitative metrics (delta and line counts), as illustrated in Figure 1.
- Statistical Power and Analysis: As requested, the "Statistical analysis of data" section has been revised. We have included a post-hoc power analysis, which confirms that our sample size of 212 athletes provides a statistical power greater than 80% to detect the observed effects, particularly for the spiral whorl (WS) pattern (Cohen’s d = 0.420). We have also clarified that a Bonferroni correction was applied for multiple comparisons in the individual finger analyses. Furthermore, we now explicitly describe the structure of the binary logistic regression model used to control for potential confounding variables.
Addressing Confounding Factors and Terminology
Reviewer 3 raised an important point regarding the potential confounding effect of retirement age and the use of the term "diagnosis."
- Age as a Confounding Factor: We acknowledge that the significant difference in retirement age between the groups could be a confounding factor. To address this, we conducted a binary logistic regression analysis, including age as a predictor. The results, now included in the revised manuscript, demonstrate that the association between the WS pattern and ACL injury history remains significant (p = 0.013) even after controlling for age. This strengthens our conclusion that the dermatoglyphic marker is an independent risk indicator.
- Terminology Adjustment: We agree that terms like "diagnosis" could be misleading. We have meticulously revised the manuscript to replace such terms with more appropriate language that reflects risk association and predisposition (e.g., "risk indicator," "indicate a higher risk," "risk assessment," "history of ACL injuries"). This change better aligns with the study's retrospective design and avoids implying causal certainty.
We believe that these comprehensive revisions have substantially improved the manuscript and have fully addressed all the insightful comments provided by the reviewers. We thank them for their time and effort, which have been invaluable in strengthening our work.
Round 2
Reviewer 1 Report
Comments and Suggestions for Authors
the author's adressed all my concerns. I have no further suggestions.
Reviewer 2 Report
Comments and Suggestions for Authors
The manuscript has improved significantly after the review process. I have no further suggestions for the authors.
Reviewer 3 Report
Comments and Suggestions for Authors
No further comments or suggestions for the authors.